# Effect of Micro-Alloyed/Alloyed Elements on Microstructure and Properties of Fe-Mn-Al-C Lightweight Steel

**Xiqiang Ren, Yungang Li, Yanfei Qi \* and Chenghao Wang**

College of Metallurgy and Energy, North China University of Science and Technology, Tangshan 063210, China; rxq0219@126.com (X.R.); liyungang59322@163.com (Y.L.); hyujik2100@163.com (C.W.)
\* Correspondence: qyf999fyq@163.com

**Abstract:** In the 14th Five Year Plan, China clearly proposes to develop the automobile strategic industry, reduce the carbon emission intensity, and formulates the carbon emission peak plan by 2030. As the automobile is the most frequently used vehicle, weight reduction can achieve the purpose of energy-saving and emission reduction and help to achieve the peak of carbon emissions as soon as possible. The lightweight automobile steel is the research hotspot in the future, and the lightweight steel has attracted much attention in the automobile manufacturing industry. Fe-Mn-Al-C lightweight steel, with its high strength, good oxidation resistance at high temperatures, good fatigue performance, high elongation, and good energy absorption during a collision, etc., has attracted the attention of researchers in the field of automotive steel. It is found that the addition of micro-alloyed/alloyed elements to Fe-Mn-Al-C lightweight steel is of great significance to improve its properties. In this paper, the effects of micro-alloyed elements (Nb and V) and alloy elements (Si, Cr, and Cu) on the microstructure, properties, and κ-carbide of Fe-Mn-Al-C lightweight steel were reviewed. The main ways of improving the properties of Fe-Mn-Al-C lightweight steel by micro-alloyed/alloyed elements were summarized and the existing problems were analyzed to provide a reference for future research.

**Keywords:** micro-alloyed elements; alloyed elements; Fe-Mn-Al-C lightweight steel



## 1. Introduction

The automobile is the product of modern civilization and one of the main means of transportation. With the intensification of the global energy crisis and environmental degradation, automobile energy consumption and exhaust emissions have attracted extensive attention. It is found that vehicle weight reduction can reduce energy consumption. Al can significantly reduce the density of steel. Adding 1 wt.% Al to the steel, the quality of the steel is reduced by 1.5% [1,2]. Among the materials used in vehicles, steel materials account for about 70% of the vehicle weight, so lightweight steel has attracted much attention in the automotive manufacturing industry. In the 14th Five Year Plan, China clearly proposes to develop the automobile strategic industry and reduce the carbon emission intensity, and formulates the carbon emission peak plan by 2030 years. Therefore, research on lightweight steel has become a strategic need for national economic construction and environmental protection. Fe-Mn-Al-C lightweight steel has the characteristics of high strength at room temperature and low temperature, good oxidation resistance at high temperature, good fatigue performance, high elongation, and good energy absorption during a collision, etc., [3–9]. According to relative contents of alloying elements (such as Al, C, and Mn) and matrix phases at high (hot working) temperatures, the Fe-Mn-Al-C lightweight steels are divided into three categories, ferritic lightweight steels, duplex lightweight steels (ferrite based duplex lightweight steels and austenite based duplex lightweight steels) and austenitic lightweight steels (see Figure 1). During the cooling process of austenitic lightweight steels, intragranular κ-carbides ((Fe,Mn)$_3$AlC precipitation

and intergranular κ-carbides ((Fe,Mn)$_3$AlC) precipitation will be produced. The κ-carbides as the main precipitate in Fe-Mn-Al-C austenite lightweight steels, the morphology, size, and distribution is directly related to the properties of lightweight steels. By adjusting heat treatment process parameters, nanometer κ-carbides can be dispersed in grains, so as to improve the comprehensive properties of lightweight steels. In order to further improve the comprehensive properties of lightweight steels, strengthening mechanisms such as grain refinement and precipitation strengthening are adopted, that is, micro-alloyed/alloyed elements are added to lightweight steels [10–13].

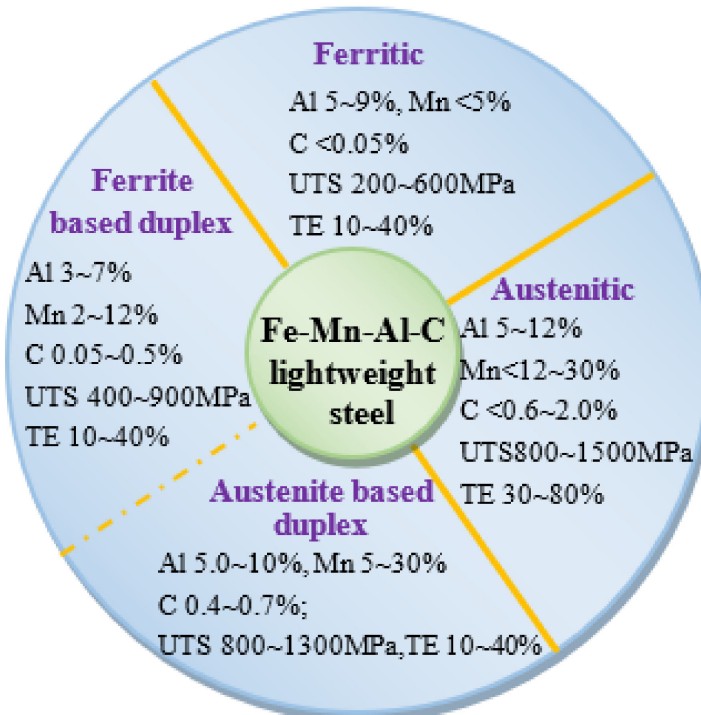

**Figure 1.** Classification of Fe-Mn-Al-C lightweight steels according to hot rolled microstructures, data from [14].

Therefore, according to the properties requirements of automotive lightweight steels, this paper selected the representative latest research results and summarized the effects of microalloying elements (Nb and V) and alloy elements (Si, Cr, and Cu) on the microstructure, properties, and κ-carbides of Fe-Mn-Al-C lightweight steels. The main ways to improve the properties of Fe-Mn-Al-C lightweight steels by micro-alloyed/alloyed elements were summarized, and the existing problems were analyzed.

## 2. Effect of Microalloyed Elements on Fe-Mn-Al-C Lightweight Steels

### 2.1. Niobium

The main functions of Nb in Fe-Mn-Al-C lightweight steels are fine grain strengthening and dispersion strengthening. The addition of Nb in Fe-Mn-Al-C lightweight steels forms inter- and intra-granular Nb(C,N). Nb(C,N) can inhibit austenite growth during ingot casting, delay recrystallization, and inhibit recrystallized grains growth during hot rolling. NbC will dissolve during high-temperature annealing and precipitate during cooling. In the deformation of metal materials, Nb(C,N) has the function of pinning dislocations. In addition, Nb acted as a suppressor of κ-carbides due to the preferential formation of NbC. Kwon et al. [15] found that adding Nb to Fe-Mn-C-Al TWIP steel can reduce its twinning dynamics and inhibits deformation twinning, and the work hardening rate of the TWIP steel is reduced due to the influence of ineffective mechanical twins. Mejía et al. [16] reported that the addition of 0.06 wt.% Nb improved the wear resistance in Fe-22Mn-1.5Si-1.5Al-0.4C steel.

The corrosion resistance of Fe-28Mn-10Al-1C steel in 3.5%NaCl solution was improved, owing to the addition of Nb to Fe-28Mn-10Al-1C steel, that is, increasing corrosion potential and polarization resistance, decreasing corrosion current density [17]. Ma et al. [18] studied the addition of 0.5% Nb to Fe-28Mn-10Al-1C austenitic lightweight steel, the austenite grain size decreased from 39.49 μm to 13.67 μm. Meanwhile, the yield strength, ultimate tensile strength, and elongation were increased by 171 MPa, 64 MPa, and 11.5%, respectively. The strength and ductility of the steel were well balanced. The precipitation of NbC inhibits dislocation slip (see Figure 2), which will enhance the precipitation strengthening effect and improve the strain hardening rate. In addition, Nb can increase the stacking fault energy of lightweight steel. However, the deformation mechanism and strengthening mechanism of steel have not changed.

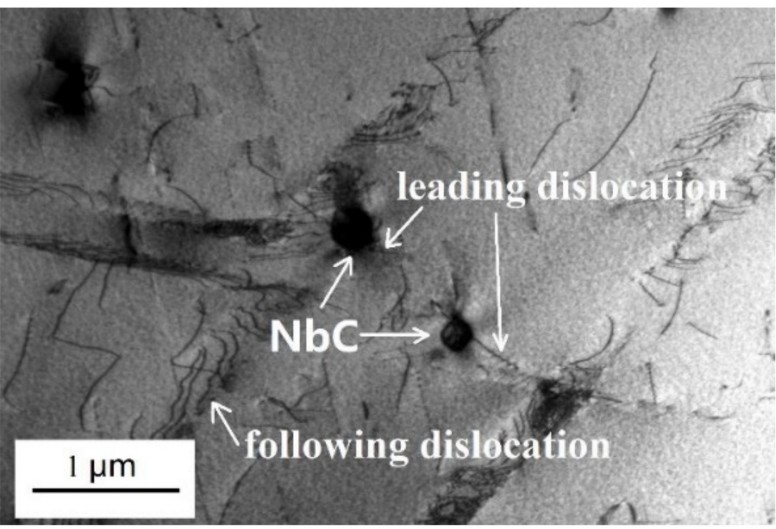

**Figure 2.** Interaction between dislocations and precipitates in 0.5Nb low-density steel at 5% strain [18].

At present, there are few reports on the addition of Nb in Fe-Mn-Al-C lightweight steel. The addition of Nb to Fe-Mn-Al-C lightweight steel can not only increase the stacking fault energy, strength, elongation, and corrosion resistance but also refine the grains. The microstructure and properties of lightweight steel are closely related to NbC particles in lightweight steel [19–21], and the research on its mechanism of improving the properties of lightweight steel needs to be deepened.

### 2.2. Vanadium

Microalloyed element V is added to steel, which can form VC particles with C in steel. Fine dispersed VC particles can improve strength and hot ductility [22]. The formation of κ-carbide and VC in austenitic steel requires solid solution C atoms in steel, so the V element can be inhibited κ-carbide formation.

Moon et al. [23] studied the microstructure and tensile property in the weld heat-affected zone of austenitic Fe-30Mn-9Al-0.9C-(0,0.5)V lightweight steel. The results showed that the tensile and yield strength of the V-added steel was improved due to grain refinement and precipitation strengthening, which is caused by the precipitation of VC particles (yellow arrows in Figure 3b). The orientation relationship between VC particles and austenite matrix is [011] γ//[011] VC (see Figure 3). In addition, the tensile strength and elongation of the welding heat-affected zone are lower than that of the base steel because of grain growth during the high-temperature thermal cycle of the heat-affected zone, but the strength of the heat-affected zone with 0.5V-added steel is higher than that V-free steel. The microstructure and mechanical properties of Fe-20Mn-9Al-1.2C-(0.6,1.0,1.4)V lightweight steels was explored by Li [24]. It is found that the comprehensive properties of 1.0V-added steel were better than other steels (0.6V and 1.4V), and a large number of nano κ-particles were precipitated in 1.0V-added steel. Although the precipitation of nano

κ-carbide particles in the steel can improve the yield strength of steel, it is difficult to improve the tensile strength because the strain hardening rate is significantly reduced. In addition, intragranular κ-carbides and $V_4C_3$ particles can be co-precipitated in the steel after cold-rolled (20%), high-temperature annealing (900 °C), and low-temperature aging (450–550 °C), and the tensile strength and specific strength of the steel were up to 1300 MPa and 190 MPa·cm$^3$/g, respectively. Therefore, intragranular nano $V_4C_3$ particles can be precipitated in lightweight steel by a V-added and appropriate heat treatment process, which improves the comprehensive properties of lightweight steel.

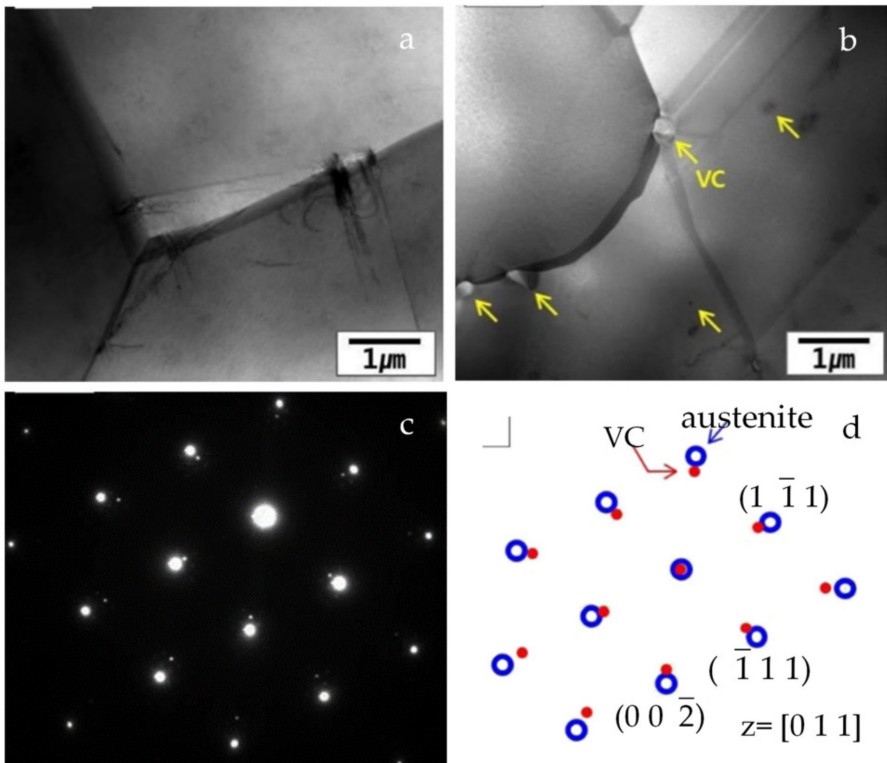

**Figure 3.** TEM micrographs of base steels (**a**) bright field image of steel 0 V, (**b**) bright field image of steel 0.5 V, (**c**) SAD pattern with Z = [011] and (**d**) computer simulated SAD pattern [23].

Adding V to lightweight steel has the effect of precipitation strengthening and grain refinement strengthening. In terms of improving the strength of lightweight steel, the V element is a more suitable additive element. However, for automotive lightweight steel, its corrosion resistance, high-temperature oxidation resistance, energy absorption during a collision, etc., should also be studied. Therefore, these properties of lightweight steel should be explored, which is of great significance to improve the comprehensive properties of Fe-Mn-Al-C lightweight steel by micro-alloyed element V. In addition, for Fe-Mn-Al-C lightweight steel with different component content, the addition amount of micro-alloyed V element and its reasonable heat treatment process parameters are the focus of research and need to be explored.

### 2.3. Co-Addition of Micro-Alloyed Elements

At present, the research on the effect of micro-alloyed elements on Fe-Mn-Al-C lightweight steel mainly focuses on single micro-alloyed elements, while the research on the effect of two or three micro-alloyed elements on it is relatively few. This part summarizes the effects of the co-addition of representative micro-alloying elements on Fe-Mn-Al-C lightweight steel.

Moon et al. [25] explored the microstructure evolution and mechanical properties of Fe-30Mn-9Al-0.9C austenitic lightweight steel containing V and Nb during aging treatment.

It is found that the strength of 0.5V-0.033Nb-added lightweight steel is higher than that of 0.5V-added lightweight steel, which is a cause of the addition of V-Nb improved the strength by grain refinement and precipitation hardening. At the beginning of aging, due to the precipitation of nano κ-carbide((Fe,Mn)$_3$AlC), the Vickers hardness increased with the extension of aging time. As the aging time is 1000~3000 min, the Vickers hardness remained stagnant. After the aging time exceeds 3000 min, the hardness increases again owing to the formation of ferrite and brittle β-Mn. The compressive behavior of Fe-27Mn-8Al-1.6C lightweight steel with 0.1Nb-0.3Mo-added at different strain rates ($10^{-3}$~$10^3$s$^{-1}$)) was investigated by Li [26]. The results showed that the microstructure of the steel after aging at 400~600 °C was austenite grains, nano-sized precipitates, (Nb,Mo)C and κ-carbides, and nano-sized precipitates and κ-carbides were evenly distributed in the matrix. With the increase in aging temperature, κ-carbide size increased, but the size and distribution of (Nb,Mo)C were not affected. Due to the enhanced interaction between dislocation and κ-carbide at a high strain rate, aging steel has a significant strain-rate strengthening effect. The strain hardening rate decreases with the increase in strain rate, which is due to more κ-carbides being sheared by dislocations leading to the reduction in the hindrance of dislocation slip at a high strain rate. Li [27] reported the effect of Ti-Mo-Nb co-addition on the microstructure and mechanical properties of Fe-26Mn-8Al-1.5C austenitic lightweight steel treated by solid solution (950 °C) and aging (500~600 °C). It was found that the steel with the co-addition of Ti-Mo-Nb has a higher yield strength because of grain refinement, more κ-carbides precipitation, and fine (Ti,Mo,Nb)C particles are distributed in the grains uniformly. However, the steel has lower ductility owing to the following two reasons: one is coarse κ-carbides with larger volume fraction, and the other one is micron-sized inter-granular (Ti,Mo,Nb)C particles along the grain boundaries.

## 3. Effect of Alloyed Elements on Fe-Mn-Al-C Lightweight Steels

### 3.1. Silicon

In lightweight steel, Si is considered to be one of the most common alloying elements, because Si can improve its fluidity, enhance the carbon activity in austenite, stabilize κ-carbide at high temperature and promote the formation of κ-carbide [26].

In Fe-30Mn-9Al-0.9C-0.5Mo lightweight steel, the addition of (0.1 wt.%) Si promotes the partitioning of carbon to κ-carbide during aging, that is, (Fe, Mn)$_3$AlC$_{0.38}$ to (Fe, Mn)$_3$AlC$_{0.51}$. In addition, Kim et al. [28] found that the addition of the Si element promoted the deformation localization, resulting in an intensively unidirectional shear band. Abedi et al. [29] found that substructure refinement has a significant effect on the strain hardening behavior of Fe-17.5Mn-8.3Al-0.74C-0.14Si lightweight steel. Fe-31.8Mn-6.09Al-1.60Si-0.40C lightweight steel showed good oxidation resistance, especially in an air environment of 600~700 °C and an oxygen environment of 600 °C [30]. Lee [31] observed the microstructure of Fe-8Al-30Mn-1.5Si-1.5C lightweight steel in a quenched state by TEM were γphase and fine (Fe,Mn)$_3$AlC phase. In the aging temperature range of 550~1000 °C, with the increase in temperature, the transformation order of quenched alloy was (Fe,Mn)$_3$AlC+DO3 → (Fe,Mn)$_3$AlC+B2 → (Fe,Mn)$_3$AlC+α → γ. This transition was not previously observed in Fe-Al-Mn-C and Fe-Al-Mn-Si-C alloys.

Wang et al. [32] investigated the formation mechanism of κ-carbides and deformation behavior of Fe-30Mn-9Al-1.2C lightweight steel with (0, 1, 2 wt.%) Si-added. The report showed that Si can significantly improve the activities of Al and C according to thermodynamic calculation. With the increase in Si content from 0 to 2 wt.%, the yield strength of steel increased from ~450 MPa to ~950 MPa, which is due to the precipitation of κ-carbides in grain interiors. The main deformation mechanism of steel is caused by the slip band in the steel, and the slip band is related to the shear of ordered nano-domains or κ-carbides. In addition, the Si-free Fe-30Mn-9Al-1.2C lightweight steel has high strain hardening, that is, the uniform distribution of these slip bands and the dynamical refinement of their spacing in the steel. However, the strain hardening of 2%Si-added Fe-30Mn-9Al-1.2C lightweight steel was decreased because of local dynamic recovery and strain softening.

The oxidation resistance of Fe-Mn-Al-C lightweight steel can be improved by adding an Si element, and the strength of lightweight steel is also improved, which is attributed to Si in the steel promoting κ-carbide precipitation. However, the high addition of Si in lightweight steel will lead to the precipitation of intergranular κ-carbide in the steel, which is unfavorable to the properties of lightweight steel. In addition, the strain hardening mechanism of lightweight steel will also change after adding the Si element.

### 3.2. Chromium Element

Cr elements can effectively improve the high-temperature oxidation resistance and environmental embrittlement resistance of materials [33,34]. In Fe-Mn-Al-C lightweight steel, Cr combines with C to form CrC, which can inhibit the formation of κ-Carbide formation in lightweight steel. There are more and more reports on the effect of Cr on Fe-Mn-Al-C lightweight steel.

Sutou et al. [35] studied the mechanical properties of Fe-20Mn-(10~14)Al-(0~1.8)C-(0,5)Cr steel and found that Fe-20Mn-11Al-1.8C-5Cr steel has the excellent properties, with a yield strength of 1040 MPa, a tensile strength of 1223 MPa and tensile elongation of 41%. Kim et al. [36] investigated the microstructure of Fe-20Mn-12Al-1.5C-(0~7)Cr lightweight steel. The results showed that 0Cr steel was composed of austenite($\gamma$) with fine intragranular κ-carbides, coarse intergranular κ-carbides, and a small amount of ferrite ($\alpha$). The microstructure of 5Cr steel is mainly composed of $\gamma$ and fine intragranular κ-carbides, a small amount of $\alpha$, and ordered phase DO3. The coarse intergranular κ-carbides disappeared and the fraction of austenite was slightly increased. It is attributed that Cr is a carbide forming element, which increases soluble C content within austenite. As Cr content exceeds 5wt.%, Cr-rich $M_7C_3$ carbides precipitate, and the fraction of ordered phase DO3 rises. The effect of Cr content on the microstructure and phase of steel is shown in Table 1.

**Table 1.** Microstructure and phase of Fe-20Mn-12Al-1.5C-xCr steel.

| Fe-20Mn-12Al-1.5C-xCr | Phase | Microstructure |
|---|---|---|
| 0 | coarse and fine κ-carbide | $\gamma$ and $\alpha$ (a small amount) |
| 2 | fine κ-carbide | $\gamma$ and $\alpha$ (a small amount) |
| 5 | DO3 and fine κ-carbide | $\Gamma$ and $\alpha$ (a small amount) |
| 5.5 | DO3, $Cr_7C_3$, and fine κ-carbide | $\gamma$ |
| 6 | DO3, $Cr_7C_3$, and fine κ-carbide | $\gamma$ |
| 7 | DO3, $Cr_7C_3$, and fine κ-carbide | $\gamma$ |

The high-temperature microstructures of Fe-9Al-30Mn-x(0.6,0.9,1.2)C-y(3,5,7)Cr lightweight steel contains ($\alpha + \gamma$), $\gamma$, and ($\alpha + \gamma + Cr_7C_3$) regions [37]. In the $\alpha$ phase, with the increase in Cr content, the phase transition sequence was $\alpha$+B2 $\rightarrow$ $\alpha$+B2+DO3 $\rightarrow$ $\alpha$ + DO3. In the $\gamma$ phase, with the increase in C content, the phase transition sequence was $\gamma \rightarrow \gamma$ + κ. The κ phase carbides (($Fe,Mn)_3AlC_x$) had an ordered $L'1_2$-type structure and the $Cr_7C_3$ precipitates had a hexagonal structure. Tsay et al. [38] studied the strength, ductility, and corrosion resistance of as-quenched Fe-28Mn-9Al-xC-1.8C lightweight steel. The results were shown in Table 2. As the Cr content was 6%, the steel exhibits a good combination of high strength, high ductility as well as corrosion resistance. The high density of fine ($Fe,Mn)_3AlC$ carbides was precipitated in the austenite matrix, and passive film of Cr and Al oxides were formed on the alloys. In addition, in a 3.5% NaCl solution, the pitting potential of Fe-28Mn-9Al-6C-1.8C lightweight steel was significantly higher than that of conventional AISI 410 martensitic stainless steel.

**Table 2.** Electrochemical parameters and mechanical properties of Fe-28Mn-9Al-xCr-1.8C, data from [38].

| Fe-28Mn-9Al-xCr-1.8C | Electrochemical Parameters | | Mechanical Properties | | |
|:---:|:---:|:---:|:---:|:---:|:---:|
| | $E_{corr}$/mV | $E_{pp}$/mV | UTS/MPa | YS/MPa | El/% |
| 0 | −846 | - | 1080 | 868 | 55.5 |
| 3 | −710 | −223 | 1092 | 876 | 47.2 |
| 5 | −571 | −65 | 1102 | 882 | 39.1 |
| 6 | −538 | −25 | 1122 | 902 | 36.5 |
| 8 | −746 | −412 | 984 | 835 | 22.6 |

Note: $E_{corr}$—corrosion potential, $E_{pp}$—pitting potential, UTS—Ultimate Tensile Strength, YS—Yield strength, El—Elongation.

Adding Cr to Fe-Mn-Al-C lightweight steel can improve its strength, high-temperature oxidation resistance, and corrosion resistance. Cr elements can inhibit the precipitation of κ-carbides in lightweight steel, and the precipitation of DO3 and rich Cr-carbides will affect the properties of lightweight steel. To obtain excellent properties of Fe-Mn-Al-C lightweight steel, it is necessary to formulate reasonable heat treatment process conditions and add an appropriate amount of Cr content. The added Cr content is related to the percentage of constituent elements of Fe-Mn-Al-C lightweight steel.

### 3.3. Copper Element

The addition of Cu in Fe-Mn-Al-C steel can expand the austenite zone. The nano Cu-rich particles precipitated in the steel can improve the corrosion resistance [39], increase fatigue crack propagation resistance [40] and improve the deformability without losing ductility [41]. The Cu element is an austenite stabilized alloy element in steel, which improves the tensile properties of steel by controlling the stacking fault energy and stability of austenite [42].

The addition of Cu to lightweight steel as an austenite stabilizer increases the volume fraction of austenite in the steel, and the recrystallization is delayed due to the solute drag effect [43]. The Fe-28Mn-9Al-0.8C-(0,1,3,5)Cu austenitic matrix lightweight steels strips were prepared in near-rapid solidification technology. It is found that the addition of 3% or 5% Cu to Fe-28Mn-9Al-0.8C lightweight steel would slightly reduce the ferrite content and change its morphology, resulting in a decrease in yield strength. After heat treatment, the yield strength of 5% Cu was higher than 3% Cu, which was because nanosized Cu-rich and κ-carbide particles were coprecipitated in the Cu-containing steel strips [44].

The hydrogen embrittlement resistance of Fe-0.8C-15Mn-7Al-(0,1,3) Cu duplex lightweight steels was evaluated by a slow-strain-rate tensile test and electrochemical H permeation test [45]. The results show that the addition of Cu to the lightweight steels increases the fraction of austenite, decreases the elongation loss, and reduces the reversible concentration and diffusivity of H, that is, the hydrogen embrittlement resistance of the lightweight steel was improved. Meanwhile, the addition of Cu promotes the formation of B2 particles, in which B2/austenite interfaces were complexly semi-coherent, and the B2 particles provided misfit dislocations at the interfaces. B2 particles promoted the transition from reversible to irreversible sites (as shown in Figure 4), which further reduced the diffusivity of H.

Adding Cu to Fe-Mn-Al-C steel can effectively improve its yield strength and alleviate hydrogen induced delayed fracture. The Fe-Cu system has a large positive mixing enthalpy. Cu and Fe are immiscible. Cu atoms can be easily separated from the supersaturated austenite matrix to form Cu-rich clusters or particles. High Cu content will provide a higher chemical driving force for the precipitation of Cu-rich particles during heat treatment [46]. Therefore, Fe-Mn-Al-C austenitic lightweight steel added with Cu has broad application prospects. Through the precipitation strengthening of fine Cu-rich particles, it can improve the strength of the steel without reducing the plasticity and alleviate the problem that it is difficult to have both strength, toughness, and plasticity. Meanwhile, the precipitation of fine Cu-rich particles can promote the precipitation of fine κ-carbide particles, which is of

great significance to improving the yield strength of lightweight steels and maintaining good plasticity.

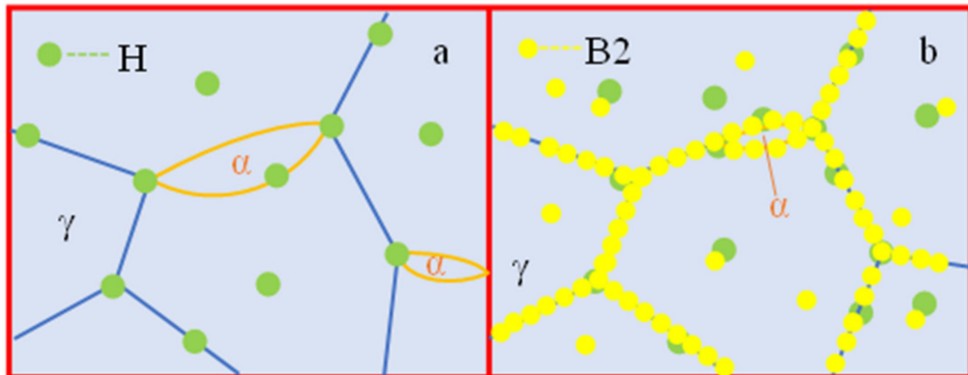

**Figure 4.** Mechanisms of microstructure change induced by concentration and diffusivity of reversible H (**a**) 0Cu and (**b**) 3Cu, adapted from [45].

## 4. Conclusions and Outlook

It is found that the addition of micro-alloy/alloy elements to Fe-Mn-Al-C lightweight steel will affect its microstructure and properties. At present, the ways to improve its properties are mainly divided into three categories: the first category is that the added micro-alloy/alloy elements combine with C or N in Fe-Mn-Al-C lightweight steel to form carbonitrides (such as Nb, V, Cr, Mo, etc.), which can play the role of fine grain strengthening and precipitation strengthening. In addition, these elements occupy the C element in the steel, which can inhibit the formation of κ-Carbide in the steel. By controlling the process parameters of rolling and heat treatment, the dispersed carbonitrides and κ-carbides can be detected in the grains, so as to improve the strength and ductility of the steel. The second type is the addition of micro-alloy/alloy elements, which can improve the carbon activity of austenite in Fe-Mn-Al-C lightweight steel, such as the Si element, promote the formation of κ-Carbides, and then play the role of precipitation strengthening. However, the size and position of κ-Carbides are not easy to control. The third type is the precipitation of added micro-alloy/alloy elements in Fe-Mn-Al-C lightweight steel (such as Cu element). The precipitation of Cu-rich particles in the steel can promote the precipitation of κ-carbide. After a reasonable heat treatment process, fine Cu-rich particles and fine κ-carbide particles can be co-precipitated in grains, thus, the comprehensive properties of lightweight steel are improved.

At this stage, in the research field of improving the properties of Fe-Mn-Al-C lightweight steel by adding micro-alloy/alloy elements, the author believes that the following problems need to be solved urgently.

(1) The changes in κ-carbide precipitation kinetics and grain size are caused by the addition of Nb, V, and Cr in Fe-Mn-Al-C lightweight steel. The research on the co-precipitation behavior and mechanism of κ-carbides and carbides of micro-alloyed/alloy elements need to be deepened, which has important guiding significance for improving the comprehensive properties of Fe-Mn-Al-C lightweight steel.

(2) The co-addition of micro-alloyed elements in Fe-Mn-Al-C lightweight steel provides a preferred method to improve the comprehensive properties of automotive steel. Microalloyed elements V-Nb and Ti-Mo-Nb have been successfully added to the steel, and the mechanical properties of the steel have been significantly improved. However, the coordination relationship between V and Nb, or Ti, Mo, and Nb are not clear concerning micro-alloyed elements and matrix steel. This is the focus and difficulty of research, solving this difficulty is of great significance for the preparation of lightweight steel with excellent comprehensive properties.

(3) Adding a Cu element to Fe-Mn-Al-C lightweight steel and equipping it with a reasonable heat treatment process can realize the co-precipitation of fine Cu-rich particles and fine κ-particles in grains. Further, it can alleviate the problem that it is difficult to have both strength and toughness and plasticity, and significantly improves the hydrogen-induced fracture properties of lightweight steel. However, the co-precipitation relationship (precipitation sequence, interdependence, interaction, etc.), and co-precipitation mechanism of Cu-rich particles and κ-carbide particles are not clear. Therefore, strengthening this research is of great significance to further improve the comprehensive properties of Fe-Mn-Al-C lightweight steel.

**Author Contributions:** Conceptualization, X.R. and Y.Q.; methodology, X.R.; software, X.R. and C.W.; validation, X.R. and C.W.; formal analysis, X.R.; investigation, X.R. and Y.Q; resources, X.R.; data curation, X.R. and Y.Q; writing—original draft preparation, X.R.; writing—review and editing, X.R., Y.L. and Y.Q; visualization, X.R.; supervision, Y.L. and Y.Q.; project administration, X.R.; funding acquisition, Y.L. All authors have read and agreed to the published version of the manuscript.

**Funding:** This research was funded by the National Natural Science Foundation of China grant number 51974129 and 52104374, Natural Science Foundation of Hebei Province grant number E2021209099.

**Institutional Review Board Statement:** Not applicable.

**Informed Consent Statement:** Not applicable.

**Data Availability Statement:** Not applicable.

**Acknowledgments:** This research was funded by the National Natural Science Foundation of China grant number 51974129 and 52104374, Natural Science Foundation of Hebei Province grant number E2021209099, Natural Science Foundation of Hebei Education Department grant number QN2020138 and Technology Innovation Team Training Plan Project grant number 21130207D.

**Conflicts of Interest:** The authors declare no conflict of interest.

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
