# Peer review of "Effect of Micro-Alloyed/Alloyed Elements on Microstructure and Properties of Fe-Mn-Al-C Lightweight Steel"

_metals, doi:10.3390/met12040695_

Round 1
Reviewer 1 Report
Xiqiang and his co-workers present in this manuscript a review of the effects of micro-alloyed elements (Nb and V) and alloy elements (Si, Cr and Cu) on the microstructure, properties and κ-carbide formation of Fe-Mn-Al-C lightweight steel. The manuscript, in general, is well-written, and the review deserves publication. The content of the manuscript fits the profile of the journal.
Comments:
--- line 202: please correct “Cr combines with C to form VC,”; VC should be CrC;
--- Table 1: please cite the original paper where data are from;
--- Figure captions, in general: proper citation to original works have to be provided, with properly citing the relevant copyright information/permission; e.g.: Reproduced with permission from [xx]. Copyright publisher, year;
--- Figure 7: provide reference and proper citing;
--- References: please check the journal’s home page and use proper reference style.
--- A final reading of the manuscript is necessary to improve the English (minor corrections).
Author Response
Comments:
--- line 202: please correct “Cr combines with C to form VC,”; VC should be CrC;
Thank you for pointing out the details. We have carefully corrected it in the manuscript. “Cr combines with C to form CrC”
--- Table 1: please cite the original paper where data are from;
Thank you for pointing out the details. We have cited the original paper where data are from in Table 2. “Table 2. Electrochemical parameters and mechanical properties of Fe-28Mn-9Al-xCr-1.8C[38].”
--- Figure captions, in general: proper citation to original works have to be provided, with properly citing the relevant copyright information/permission; e.g.: Reproduced with permission from [xx]. Copyright publisher, year;
--- Figure 7: provide reference and proper citing;
Thank you for pointing out the details. We have modified it in the manuscript. “Figure 4(original Figure7). Mechanisms of microstructure change induced by concentration and diffusivity of reversible H (a) 0Cu and (b) 3Cu.”
Figure.4 Mechanisms of microstructure change induced by concentration and diffusivity of re-versible H (a) 0Cu and (b) 3Cu.
--- References: please check the journal’s home page and use proper reference style.
Thank you for your careful work. We have checked the journal’s home page and modifiied reference style.
--- A final reading of the manuscript is necessary to improve the English (minor corrections).
Thank you for pointing out the question. We have carefully corrected the language of manuscript.

Reviewer 2 Report
General comment:
The text has lots of interesting information.
Nonetheless, it is difficult to have a good understanding of this field due to somewhat disorganised or less linked scientific results.
This article is a review. But some very important references are missing, for example:
ZAMBRANO, O. A. A general perspective of Fe–Mn–Al–C steels. Journal of materials science, v. 53, n. 20, p. 14003-14062, 2018.
Howell, Ryan A., and David C. Van Aken. "A literature review of age hardening Fe-Mn-Al-C alloys." (2009): 193.
Some paragraphs that need improvement:
“According to hot rolled 36 microstructure, the matrix phase of Fe-Mn-Al-C lightweight steels can be divided into 37 three categories, ferrite lightweight steels, duplex lightweight steels (ferrite based duplex 38 lightweight steels and austenite based duplex lightweight steels) and austenite light-39 weight steels (see Fig. 1).”
comments: This information out of context.
“In order to further improve the properties of lightweight steels, researchers consider adding microalloyed/alloyed elements to lightweight steels, using a variety of strengthening mechanisms such as grain refinement and precipitation strengthening to improve the comprehensive properties of the material, and achieved remarkable results.”
suggestion: enter a reference
“Therefore, according to the properties requirements of automotive lightweight steels, this paper selected the representative latest research results and summarized the effects of microalloyed elements (Nb and V) and alloy elements (Si, Cr and Cu) on the microstructure, properties and κ-carbides of Fe-Mn-Al-C lightweight steels. The main ways of 54 improving the properties of Fe-Mn-Al-C lightweight steels by microalloyed/alloyed elements were summarized, and the existing problems were analyzed.”
comments:
- Effect of microalloyed elements on Fe-Mn-Al-C lightweight steels
- Effect of alloyed elements on Fe-Mn-Al-C lightweight steels
question: What is the difference? explain better.
“At present, there are few reports on the addition of Nb in Fe-Mn-Al-C lightweight 86 steel. The addition of Nb to Fe-Mn-Al-C lightweight steel can not only increase the stacking fault energy, strength, elongation and corrosion resistance, but also refine the grains. The microstructure and properties of lightweight steel is closely related to NbC particles in lightweight steel, the research on its mechanism of improving the properties of light weight steel needs to be deepened.”
suggestion: enter a reference
Suggestion: present in the form of tables, differentiating the effects on the microstructure, formation of phases of the effects on mechanical properties and corrosion resistance.
question: what gaps need research?
The conclusion is too great.
“(4) It is necessary to strengthen the research on the microstructure behavior of Fe- Mn-Al-C lightweight steel with microalloyed/alloyed elements during tension and compression, especially the behavior and mechanism of the carbides of microalloyed/alloying elements and κ-carbides in this process. It has important guiding significance for mastering the microstructure and properties of automobile steel in the process of forming and collision.”
question: This information out of context. This information has not been previously displayed.
Author Response
Reviewer 2:
General comment:
The text has lots of interesting information.
Nonetheless, it is difficult to have a good understanding of this field due to somewhat disorganised or less linked scientific results.
Thank you for your meaningful advice. We revised the manuscript and linked scientific results
This article is a review. But some very important references are missing, for example:
ZAMBRANO, O. A. A general perspective of Fe–Mn–Al–C steels. Journal of materials science, v. 53, n. 20, p. 14003-14062, 2018.
Howell, Ryan A., and David C. Van Aken. "A literature review of age hardening Fe-Mn-Al-C alloys." (2009): 193.
Thank you for your valuable advice. We carefully read these two literatures and cited them (Reference [3,4]). In addition, we have added reading and citation of relevant literatures[5-9].
Reference:
[3] Zambrano, O. A. A general perspective of Fe–Mn–Al–C steels. J. Mater. Sci., 2018, 53, 14003-14062. https://doi.org/10.1007/s10853-018-2551-6
[4] Howell, R. A.; Van Aken, D.C. A literature review of age hardening Fe-Mn-Al-C alloys. Iron and Steel Technology, 2009, 6,193-212. https://scholarsmine.mst.edu/matsci_eng_facwork/1283/
[5] Lai, Z. H.; Sun, Y. H.; Lin, Y. T.; Tu, J. F.; Yen, H. W. Mechanism of twinning induced plasticity in austenitic lightweight steel driven by compositional complexity. Acta Mater., 2021, 210, 116814. https://doi.org/10.1016/j.actamat.2021.116814
[6] Ley, N. A.; Young, M. L.; Hornbuckle, B. C.; Field, D. M.; Limmer, K. R. Toughness enhancing mechanisms in age hardened Fe–Mn–Al–C steels. Mat. Sci. Eng. A, 2021, 820, 14151. https://doi.org/10.1016/j.msea.2021.141518
[7] Ren, P.; Chen, X. P.; Cao, Z. X.; Mei. L.; Li, W. J.; Cao, W. Q.; Liu, Q. Synergistic strengthening effect induced ultrahigh yield strength in lightweight Fe-30Mn-11Al-1.2C steel. Mat. Sci. Eng. A, 2019, 752, 160-166. https://doi.org/10.1016/j.msea.2019.03.006
[8] Hwang, J. H.; Trang, T. T. T.; Lee, O.; Park, G.; Zargaran, A.; Kim, N. J. Improvement of strength – ductility balance of B2-strengthened lightweight steel. Acta Mater., 2020, 191, 1-12. https://doi.org/10.1016/j.actamat.2020.03.022
[9] Lee, K. Park, S. J.; Lee, J.; Moon, J.; Kang, J. Y.; Kim, D. I.; Suh, J. Y.; Han, H. N. Effect of aging treatment on microstructure and intrinsic mechanical behavior of Fe–31.4Mn–11.4Al–0.89C lightweight steel. J. Alloy. Compd. 2016, 656, 805-811. https://doi.org/10.1016/j.jallcom.2015.10.016
Some paragraphs that need improvement:
“According to hot rolled 36 microstructure, the matrix phase of Fe-Mn-Al-C lightweight steels can be divided into 37 three categories, ferrite lightweight steels, duplex lightweight steels (ferrite based duplex 38 lightweight steels and austenite based duplex lightweight steels) and austenite light-39 weight steels (see Fig. 1).”
comments: This information out of context.
Thank you for your question and valuable suggestion. We relearned the original article [14] and revised the manuscript.
This part shows that Fe-Mn-Al-C lightweight steels is divided into three categories, ferrite lightweight steels, duplex lightweight steels (ferrite based duplex lightweight steels and austenite based duplex lightweight steels) and austenite lightweight steels.
“According to relative contents of alloying elements (such as Al, C and Mn) and matrix phases at high (hot working) temperatures, the Fe-Mn-Al-C lightweight steels is divided into three categories, ferritic lightweight steels, duplex lightweight steels (ferrite based duplex lightweight steels and austenite based duplex lightweight steels) and austenitic lightweight steels (see Fig. 1).”
[14] Chen, S. P.; Rana, R.; Haldar, A.; Ray, R.K. Current state of Fe-Mn-Al-C low density steels. Prog. Mater. Sci. 2017, 89, 345-391. https://doi.org/10.1016/j.pmatsci.2017.05.002
“In order to further improve the properties of lightweight steels, researchers consider adding microalloyed/alloyed elements to lightweight steels, using a variety of strengthening mechanisms such as grain refinement and precipitation strengthening to improve the comprehensive properties of the material, and achieved remarkable results.”
suggestion: enter a reference
Thank you for your valuable advice. We entered references.
“In order to further improve the comprehensive properties of lightweight steels, strengthening mechanisms such as grain refinement and precipitation strengthening are adopted, that is, micro alloyed/alloyed elements are added to lightweight steels [10-13].’’
[10] Moon, J.; Park, S. J.; Jang, J. H.; Lee, T. H.; Lee, C. H.; Hong, H. U.; Han, H. N.; Lee J.; Lee, B. H.; Lee, C. Investigations of the microstructure evolution and tensile deformation behavior of austenitic Fe-Mn-Al-C lightweight steels and the effect of Mo addition. Acta Mater., 2018, 147, 226-235. https://doi.org/10.1016/j.actamat.2018.01.051
[11] Park, G.; Nam, C.H.; Zargaran, A.; Kim, N. J. Effect of B2 morphology on the mechanical properties of B2-strengthened lightweight steels. Scripta Mater., 2019, 165,68-72. https://doi.org/10.1016/j.scriptamat.2019.02.013
[12] Park, B. H.; Kim, C.W.; Lee, K. W.; Park, J. U.; Park, S. J.; Hong, H. U. Role of Nb Addition on Microstructural Stability and Deformation Behaviors of FeMnAlC Lightweight Steels at 400℃. Metall. Mater. Trans. A. 2021, 52, 4191-4205. https://doi.org/10.1007/s11661-021-06379-2.
[13] Gutierrez-Urrutia I. Low Density Fe–Mn–Al–C Steels: Phase Structures, Mechanisms and Properties. ISIJ International, 2021, 61, 16-25. https://doi.org/10.2355/isijinternational.ISIJINT-2020-467
“Therefore, according to the properties requirements of automotive lightweight steels, this paper selected the representative latest research results and summarized the effects of microalloyed elements (Nb and V) and alloy elements (Si, Cr and Cu) on the microstructure, properties and κ-carbides of Fe-Mn-Al-C lightweight steels. The main ways of 54 improving the properties of Fe-Mn-Al-C lightweight steels by microalloyed/alloyed elements were summarized, and the existing problems were analyzed.”
comments:
- Effect of microalloyed elements on Fe-Mn-Al-C lightweight steels
- Effect of alloyed elements on Fe-Mn-Al-C lightweight steels
question: What is the difference? explain better.
Thank you for your question. According to the content of added elements in steel, it can be divided into microalloyed elements and alloyed elements. The content of microalloyed elements is lower than that of alloying elements. In addition, These microalloyed elements and alloyed elements are universally acknowledged in the steel industry.
“At present, there are few reports on the addition of Nb in Fe-Mn-Al-C lightweight 86 steel. The addition of Nb to Fe-Mn-Al-C lightweight steel can not only increase the stacking fault energy, strength, elongation and corrosion resistance, but also refine the grains. The microstructure and properties of lightweight steel is closely related to NbC particles in lightweight steel, the research on its mechanism of improving the properties of light weight steel needs to be deepened.”
suggestion: enter a reference
Thank you for your valuable advice. We entered references.
“At present, there are few reports on the addition of Nb in Fe-Mn-Al-C lightweight steel. The addition of Nb to Fe-Mn-Al-C lightweight steel can not only increase the stacking fault energy, strength, elongation and corrosion resistance, but also refine the grains [20,21]. The microstructure and properties of lightweight steel is closely related to NbC particles in lightweight steel, the research on its mechanism of improving the properties of light weight steel needs to be deepened.”
[20] Zhou, N. P.; Song, R. B.; Song, R. F.; Li, X.; Li, J. J. Influence of Nb Addition on Microstructure and Mechanical Properties of Medium-Mn Low-Density Steels. Steel Research International, 2018, 89, 1700552. https://doi.org/10.1002/srin.201700552
[21] Han, K. H. The microstructures and mechanical properties of an austenitic Nb-bearing Fe–Mn–Al–C alloy processed by controlled rolling. Materials Science and Engineering A, 2000, 279, 1–9. https://doi.org/10.1016/S0921-5093(99)00690-5
Suggestion: present in the form of tables, differentiating the effects on the microstructure, formation of phases of the effects on mechanical properties and corrosion resistance.
Thank you for your meaningful advice. The microstructure and properties of Fe-Mn-Al-C lightweight steel is related to the content of Mn, Al and C. In the literatures, the contents of Mn, Al and C in Fe-Mn-Al-C lightweight steels are inconsistent. Therefore, it is not convincing to compare the effects of microalloyed/alloyed elements on microstructure and properties.
question: what gaps need research?
The conclusion is too great.
“(4) It is necessary to strengthen the research on the microstructure behavior of Fe- Mn-Al-C lightweight steel with microalloyed/alloyed elements during tension and compression, especially the behavior and mechanism of the carbides of microalloyed/alloying elements and κ-carbides in this process. It has important guiding significance for mastering the microstructure and properties of automobile steel in the process of forming and collision.”
question: This information out of context. This information has not been previously displayed.
Thank you for your careful work. We modified conclusion (1-3) and deleted conclusion (4).
(1) The changes of κ-carbide precipitation kinetics and grain size caused by the addition of Nb, V and Cr in Fe-Mn-Al-C lightweight steel. The research on the co-precipitation behavior and mechanism of κ-carbides and carbides of microalloyed/alloy elements needs to be deepened, which has important guiding significance for improving the comprehensive properties of Fe-Mn-Al-C lightweight steel.
(2) The co-addition of microalloyed elements in Fe-Mn-Al-C lightweight steel provides a preferred method to improve the comprehensive properties of automotive steel. Microalloyed elements V-Nb and Ti-Mo-Nb have been successfully added to the steel, and the mechanical properties of the steel have been significantly improved. However, the coordination relationship between V and Nb, or Ti, Mo and Nb is not clear, so as to microalloyed elements and matrix steel. This is the focus and difficulty of research, solving this difficulty is of great significance for the preparation of lightweight steel with excellent comprehensive properties.
(3) Adding Cu element to Fe-Mn-Al-C lightweight steel and equipped with reasonable heat treatment process can realize the co-precipitation of fine Cu rich particles and fine κ-particles in grains. Alleviate the problem that it is difficult to have both strength and toughness and plasticity, and significantly improves the hydrogen induced fracture properties of lightweight steel. However, the co-precipitation relationship (precipitation sequence, interdependence and interaction etc.) and co-precipitation mechanism of Cu rich particles and κ-carbide particles is not clear. Therefore, strengthening this research is of great significance to further improve the comprehensive properties of Fe-Mn-Al-C lightweight steel.

Reviewer 3 Report
- This review manuscript mainly deals with the effects of alloying elements on microstructures and properties of Fe-Mn-Al-C steel. However, the references cited are not comprehensive nor instructive for readers unfamiliar to this field. If one consults the google scholar, for example, he will found many review articles and excellent papers. It is strange that the references cited by the authors are almost by Chinese people.
- Total seven figures and one table have been adapted from references by the other research groups. Where is the authors' original contribution.
- The density of Fe-Mn-Al-C steel versus Al content should be given. This is indispensable for this review entitles "lightweight steel".
Author Response
Reviewer 3:
- This review manuscript mainly deals with the effects of alloying elements on microstructures and properties of Fe-Mn-Al-C steel. However, the references cited are not comprehensive nor instructive for readers unfamiliar to this field. If one consults the google scholar, for example, he will found many review articles and excellent papers. It is strange that the references cited by the authors are almost by Chinese people.
Thank you for pointing out the question. We have revised the references cited [3-9]. In addition, we changed some Chinese paper into English paper.
[3]. Zambrano, O. A. A general perspective of Fe–Mn–Al–C steels. Journal of materials science, 2018, 53, 14003-14062. https://doi.org/10.1007/s10853-018-2551-6
[4] Howell, R. A.; Van Aken, D.C. A literature review of age hardening Fe-Mn-Al-C alloys. Iron and Steel Technology, 2009, 6,193-212. https://scholarsmine.mst.edu/matsci_eng_facwork/1283/
[5] Lai, Z. H.; Sun, Y. H.; Lin, Y. T.; Tu, J. F.; Yen, H. W. Mechanism of twinning induced plasticity in austenitic lightweight steel driven by compositional complexity. Acta Mater., 2021, 210, 116814. https://doi.org/10.1016/j.actamat.2021.116814
[6] Ley, N. A.; Young, M. L.; Hornbuckle, B. C.; Field, D. M.; Limmer, K. R. Toughness enhancing mechanisms in age hardened Fe–Mn–Al–C steels. Mat. Sci. Eng. A, 2021, 820, 14151. https://doi.org/10.1016/j.msea.2021.141518
[7] Ren, P.; Chen, X. P.; Cao, Z. X.; Mei. L.; Li, W. J.; Cao, W. Q.; Liu, Q. Synergistic strengthening effect induced ultrahigh yield strength in lightweight Fe-30Mn-11Al-1.2C steel. Mat. Sci. Eng. A, 2019, 752, 160-166. https://doi.org/10.1016/j.msea.2019.03.006
[8] Hwang, J. H.; Trang, T. T. T.; Lee, O.; Park, G.; Zargaran, A.; Kim, N. J. Improvement of strength – ductility balance of B2-strengthened lightweight steel. Acta Mater., 2020, 191, 1-12. https://doi.org/10.1016/j.actamat.2020.03.022
[9] Lee, K. Park, S. J.; Lee, J.; Moon, J.; Kang, J. Y.; Kim, D. I.; Suh, J. Y.; Han, H. N. Effect of aging treatment on microstructure and intrinsic mechanical behavior of Fe–31.4Mn–11.4Al–0.89C lightweight steel. J. Alloy. Compd. 2016, 656, 805-811. https://doi.org/10.1016/j.jallcom.2015.10.016
- Total seven figures and one table have been adapted from references by the other research groups. Where is the authors' original contribution.
Thank you for your question. Figures and tables are only for readers to understand the content. We summarize the microstructure and phase of Fe-20Mn-12Al-1.5C-xCr steel, which is listed in Table 1.
Table 1. Microstructure and phase of Fe-20Mn-12Al-1.5C-xCr steel
|
Fe-20Mn-12Al- 1.5C-xCr |
Phase |
Microstructure |
|
0 |
coarse and fine κ-carbide |
γ and α (a small amount) |
|
2 |
fine κ-carbide |
γ and α (a small amount) |
|
5 |
DO3 and fine κ-carbide |
γ and α (a small amount) |
|
5.5 |
DO3, Cr7C3 and fine κ-carbide |
γ |
|
6 |
DO3, Cr7C3 and fine κ-carbide |
γ |
|
7 |
DO3, Cr7C3 and fine κ-carbide |
γ |
- The density of Fe-Mn-Al-C steel versus Al content should be given. This is indispensable for this review entitles "lightweight steel".
Thank you for your meaningful question. Your question is worth exploring. The density of Fe-Mn-Al-C steel is related to the content of Mn, Al and C. Some articles report the density of Fe-Mn-Al-C steel. Adding 1wt.% Al to the steel, the quality of the steel is reduced by 1.5% [1-2]. This information is added in the introduction of the paper.
However, the study on the density of Fe-Mn-Al-C steel with only the change of Al content does not constitute a system. But the question is very good. We will study it next.
[1] Frommeyer, G.; Bruex, U. Microstructures and mechanical properties of high-strength Fe-Mn-Al-C light-weight TRIPLEX steels. Steel Res. Int., 2006, 77, 627-633. https://doi.org/10.1002/srin.200606440
[2] Sohn, S. S.; Song, H.; Suh, B.-C.; Kwak, J. -H., Lee, B. -J.; Kim, N. J.; Lee, S. Novel ultra-high-strength (ferrite+austenite) duplex lightweight steels achieved by fine dislocation substructures (Taylor lattices), grain refinement, and partial recrystallization. Acta Mater., 2015, 96, 301-310. https://doi.org/10.1016/j.actamat.2015.06.024

Round 2
Reviewer 2 Report
The current version was much more interesting.